# Biological data questions the support of the self inhibition required for pattern generation in the half center model

**Matthias Kohler**[1]*, **Philipp Stratmann**[1,2], **Florian Röhrbein**[1], **Alois Knoll**[1], **Alin Albu-Schäffer**[1,2], **Henrik Jörntell**[3]

1 Department of Informatics, Technical University of Munich, Garching, Germany, 2 German Aerospace Center, Institute of Robotics and Mechatronics, Weßling, Germany, 3 Department of Experimental Medical Science, Lund University, Lund, Sweden

* kohler@in.tum.de

## Abstract

Locomotion control in mammals has been hypothesized to be governed by a central pattern generator (CPG) located in the circuitry of the spinal cord. The most common model of the CPG is the half center model, where two pools of neurons generate alternating, oscillatory activity. In this model, the pools reciprocally inhibit each other ensuring alternating activity. There is experimental support for reciprocal inhibition. However another crucial part of the half center model is a self inhibitory mechanism which prevents the neurons of each individual pool from infinite firing. Self-inhibition is hence necessary to obtain alternating activity. But critical parts of the experimental bases for the proposed mechanisms for self-inhibition were obtained *in vitro*, in preparations of juvenile animals. The commonly used adaptation of spike firing does not appear to be present in adult animals *in vivo*. We therefore modeled several possible self inhibitory mechanisms for locomotor control. Based on currently published data, previously proposed hypotheses of the self inhibitory mechanism, necessary to support the CPG hypothesis, seems to be put into question by functional evaluation tests or by *in vivo* data. This opens for alternative explanations of how locomotion activity patterns in the adult mammal could be generated.

## Introduction

Decerebrated animals are capable of locomotion, which can be initiated by propelling a treadmill on which the animal stands [1, 2]. Even a spinal cord isolated from its muscles and sensory afferents is capable of generating activity resembling the activity observed in an intact spinal cord during locomotion [3, 4]. This is called *fictive locomotion*, which can be initiated by electrical stimulation of the brainstem. To explain observations such as these, the concept of the central pattern generator (CPG) was proposed [1]. The CPG was envisioned to be a neuronal circuit or neuronal element located in the spinal cord that can generate oscillatory output without oscillatory input.

**Data Availability Statement:** All relevant data are within the paper.

**Funding:** This research received funding from the German Aerospace Center under the research and

development contracts D / 572 / 67 24 68 70, D / 572 / 67 27 00 80 and from the European Union's Horizon 2020 Framework Programme for Research and Innovation under the Specific Grant Agreement No. 720270 (Human Brain Project SGA1) and the Specific Grant Agreement No. 785907 (Human Brain Project SGA2).

**Competing interests:** The authors have declared that no competing interests exist.

**Fig 1. The half center model.** Each half center is formed by a pool of neurons and symbolized by a node. Both centers are reciprocally coupled by inhibitory synapses and receive a constant stimulation $Ex_{in}$.

A possible neural circuit that may underlie the CPG was also proposed [1], called the half center model, which is illustrated in Fig 1. Two pools of neurons, which correspond to the half centers, would reciprocally inhibit each other. Both half centers would be driven by a constant excitatory drive. The reciprocal inhibition would ensure that high activity in one center leads to low activity in the other center, leading to alternating oscillation in activity between the two half centers. The alternation would ensure left right alternation and alternation between antagonists. This model would explain locomotion in decerebrated animals [1], fictive locomotion [5] and locomotion patterns in intact animals [6]. Neurophysiological evidence for a circuit fitting the half center model was found [7] and subsequently substantiated by further findings [8–10]. Furthermore, in the adult spinal cord inhibition can be maintained reliably also at high activation rates of the inhibitory interneurons [11, 12]. Hence effects where inhibitory synapses would be excitatory, as sometimes suggested for the juvenile [13–15], are not an issue in the context modeled here. The half center is a key component of computational and other models of the CPG [5, 6, 16–19]. Asymmetric half center models have been proposed to explain alternation between flexor and extensor [20]. We focus on the symmetric case. Experimental studies show that reciprocal inhibition is necessary to ensure left right alternation [21], hence reciprocal inhibition must be present in the CPG.

However, it was already noted that two groups of neurons inhibiting each other are not sufficient to generate oscillation [1, 7]. We will formally demonstrate why this is the case using a mean field model. Without a mechanism that stops neurons from indefinite activity, one half center will stay active without allowing the other half center to become active. Hence, in the half center model, self inhibition with a variable amount of delay is an additional requirement for the generation of alternating oscillations to be able to explain locomotion at variable frequencies. This notion of self inhibition is consistent with previous work and proposed CPG mechanisms [1, 7, 22, 23]. The scope of the present paper is to provide an overview of possible self inhibitory mechanisms and discuss their suitability for locomotor control.

There are several possible self inhibitory mechanisms. Persistent inward currents are widely considered as such a mechanism in models of spinal CPGs [6, 19, 23–27]. These would manifest themselves either as spike frequency adaptation or bursting behavior [22]. There is experimental support for this mechanism, however often coming from spinal cord preparations of juvenile or newborn spinal cord. In contrast spinal neurons in adult mammals *in vivo* show no spike frequency adaptation on a timescale or of a magnitude relevant to locomotion [28]. Instead, their firing rate has a simple linear relationship to the amount of excitatory input. The

linear model of the spike generation mechanism generated by these data, accurately captures the spike firing responses recorded under fictive locomotion as reported by other authors [28–30] in adult *in vivo* dorsal and ventral spinocerebellar tract neurons, which are part of the spinal interneuron population [31, 32]. This observation supports the notion that in the adult spinal cord neuronal firing patterns are dominated by synaptic inputs rather than intrinsic conductances, and hence that a linear firing rate model accurately predicts the neuronal spike responses. Also in *in vitro* studies of adult spinal cord tissue [33], motoneurons have constant input-output relationships over many seconds, and even though small spike firing adaptation can be observed it is orders of magnitudes smaller than that which can be observed in the juvenile slice. Hence a persistent inward current of a magnitude that would be sufficient to gradually reduce neural firing rates to the extent that would cause a decrease in inhibition in the opposing half center does not appear to be in effect in adult mammals *in vivo*. A likely reason is that neurons in vivo have a much higher background synaptic conductance, which effectively shunt out prominent activation of inherent conductances that may still be inherent in the membrane of the neuron at adult age. This phenomenon has been demonstrated for the deep cerebellar nuclear neurons [34], which have prominent intrinsic conductances in juvenile brain slices, but there intrinsic conductances are normally not activated *in vivo*. Hence from existing observations in adult mammals *in vivo*, it would seem that the spinal cord circuitry would have to rely on other self inhibitory mechanisms than spike frequency adaptation or intrinsically generated bursting, and our study assumes this scenario to apply.

If cellular level properties cannot explain self-inhibition, other possible self-inhibitory mechanisms can be found in the temporal properties of the synapses and the network structures formed by such synaptic connections. One possible candidate is synaptic short term depression (STD) to repetitive synaptic activation. For the STD to be able to create alternating oscillatory output, it needs to be at the center of two circuitry components connected by reciprocal inhibition, just like in the classical half center model. This mechanism has been proposed as the self-inhibitory mechanism to explain alternating oscillation in the spinal cord of the lamprey [35]. A natural extension of this mutual inhibition network is to provide each half center with a network structure to gradually accumulate activity over time, so that the half center becomes inhibited after a certain amount of time depending on the level of excitation that is provided to the network. As either of these two circuitry structures would have to form the foundation for any circuitry model implementing the half center hypothesis, we here decided to simulate these two network scenarios to explore what frequencies of alternating oscillations they would be able to support, and under what conditions with respect to the synaptic time constants.

Whereas our simulations show that both of these mechanisms could be made to work to produce alternating oscillations, these cover only part of the frequency range observed for locomotion and moreover require parameter settings for synaptic decay time constants and recovery time constants from STD, which can be observed in some *in vitro* preparations of juvenile spinal cord but that appear unlikely to be present *in vivo*. These observations put the probability of that the half center model could work in the adult mammalian spinal cord *in vivo* into question.

## Results

This section starts out by proving that a self-inhibitory mechanism is required for pattern generation in the half center model. We will do so by using a mean field model. Then we simulate two different self-inhibitory mechanisms in the mammalian spinal cord. We examine the two envisioned mechanisms by simulations with neuron models whose parameter values are based

on measured data from spinal interneurons [36–40], to find out if they are suitable for generating locomotion with the half center model. For a self-inhibitory mechanism to be useful in pattern generation for locomotion it must be capable to generate patterns with frequencies in the range of locomotion. This range is often considered to be 1 Hz to 10 Hz [41]. Neurons are modeled using the leaky integrate and fire model, where each neuron is modeled as a capacitor in parallel to a resistor (single compartment), together with a mechanism that resets the membrane voltage each time it crosses a spiking threshold. This allows the precise modeling of the timescale on which the neural mechanisms act. Leaky integrate fire neuron models have previously been used to model the timescales of neural mechanisms [42].

## Self inhibition is required for pattern generation

The half center model without self-inhibition can be formulated as a mean field model, similar to previous approaches [42]. Here we will use the mean field model to show that oscillation cannot occur without self-inhibition. In the mean field model the dynamics of the instantaneous firing rate of each of the two half centers is modeled. In a simulation of the network in Fig 1 let $v_i: \mathbb{R} \rightarrow \mathbb{R}$, $i \in \{1, 2\}$ be the instantaneous firing rates of the half centers at each point in time. The half centers are coupled to each other with inhibitory weights $w_1, w_2 \in \mathbb{R}^-$. Each half center receives an excitatory drive $Ex_{in,i} \in \mathbb{R}^+$, $i \in \{1, 2\}$. High activity in one half center suppresses the activity in the opposing half center. Hence the dynamics of the firing rates can be modeled with the following differential equation

$$\frac{d}{dt} v(t) = \begin{pmatrix} 0 & w_1 \\ w_2 & 0 \end{pmatrix} v(t) + Ex_{in},$$

with initial value $v(0) = v_0 \in \mathbb{R}^2$ and where $t \in \mathbb{R}^+$ is time. Homogenizing the equation gives

$$\frac{d}{dt} \begin{pmatrix} v_1(t) \\ v_2(t) \\ Ex_{in,1}(t) \\ Ex_{in,2}(t) \end{pmatrix} = \begin{pmatrix} 0 & w_1 & 1 & 0 \\ w_2 & 0 & 0 & 1 \\ 0 & 0 & 0 & 0 \\ 0 & 0 & 0 & 0 \end{pmatrix} \begin{pmatrix} v_1(t) \\ v_2(t) \\ Ex_{in,1}(t) \\ Ex_{in,2}(t) \end{pmatrix},$$

with initial values $v(0) = v_0$ and $Ex_{in,i}(0) = Ex_{in,i}$. The eigenvalues of the system matrix are $\lambda_{1,2} = \pm\sqrt{w_1 w_2}$ and $\lambda_{3,4} = 0$. The system does not oscillate because the eigenvalues contain no imaginary part. In fact, giving each half center $i$ self-inhibition proportional to its activity with weight $w_i' \in \mathbb{R}^-$, $i \in \{1, 2\}$ is not sufficient to generate oscillation. To show this, the situation can be modeled by modifying the previous model

$$\frac{d}{dt} \begin{pmatrix} v_1(t) \\ v_2(t) \\ Ex_{in,1}(t) \\ Ex_{in,2}(t) \end{pmatrix} = \begin{pmatrix} w_1' & w_1 & 1 & 0 \\ w_2 & w_2' & 0 & 1 \\ 0 & 0 & 0 & 0 \\ 0 & 0 & 0 & 0 \end{pmatrix} \begin{pmatrix} v_1(t) \\ v_2(t) \\ Ex_{in,1}(t) \\ Ex_{in,2}(t) \end{pmatrix}.$$

The eigenvalues of the system matrix are $\lambda_{1,2} = \frac{w_1' + w_1' \pm \sqrt{(-w_1' - w_2')^2 - 4(w_1' w_2' - w_1 w_2)}}{2}$ and $\lambda_{3,4} = 0$. The eigenvalues contain no complex part, hence the system is not oscillatory.

## Short term depression

To examine if short term depression (STD) of synaptic transmission could be a suitable self-inhibitory mechanism in the half center model, we explored the oscillatory patterns that can be generated by STD. In synapses that are repetitively activated STD can occur, presumably due to overload of the molecular machinery for the presynaptic release of synaptic vesicles or changes in the postsynaptic responsiveness [43, 44] and has been observed essentially everywhere in the central nervous system [36, 43, 45, 46]. Since different studies have reported widely different results on the properties of STD across different brain areas, we studied the effects of STD on the half center model across a wide range of time constants of recovery from STD, as well as time constants of the synaptic decay. These parameters (Table 1) were based on experimental data from spinal interneurons [36, 37]. The simulated network was the half center model shown in Fig 1.

For the simulations we used a synapse model which simply scales down the synaptic weight in accordance with the time constant of recovery from STD $\tau_{rec}$ and length of the interval to the previous spike [47–49]. It is necessary to fit the model parameter $\tau_{rec}$ to measured data, which we will describe below.

In the vertebrate spinal cord, short term depression has been studied [36]. In this study, presynaptic stimulation is given at a certain frequency. In the postsynaptic neuron, the amplitudes of the postsynaptic potentials are measured. The depression is measured as the fraction of the minimum postsynaptic potential relative to the first postsynaptic potential. We fit the time constant of recovery from STD $\tau_{rec}$ by recreating the experimental observation from this particular study [36] in a simulation. In this study, the neurons were stimulated at 10 Hz. The postsynaptic potentials induced by each stimulation pulse decay nearly to resting potential. The average depression at 10 Hz stimulation is $\mu = 71\%$, with a standard error of $S.E. = 4\%$. We use $\mu - 2 * S.E. = 63\%$ and $\mu + 2 * S.E. = 79\%$ as cutoff values for realistic depressions. In a simulation separate from the simulation of the half center model we simulated two neurons, the

**Table 1. This table lists all parameters of the neuron and synapse models as used in the specific simulations.** The neuron parameters are common in all simulation. In case one parameter was varied in different simulations a range and step size are given. If parameter was constant in all simulations only its value is given.

| Parameter | Description | Value or Range | Step Size |
|---|---|---|---|
| Neuron Model | Leaky integrate and fire neuron | `iaf_psc_alpha` | |
| $E_L$ | Resting membrane potential | −70.0 mV | |
| $C_m$ | Membrane capacitance | 45.0 pF | |
| $\tau_m$ | Membrane time constant | 55.0 ms | |
| $V_{th}$ | Spike threshold | −55.0 mV | |
| $t_{ref}$ | Refractory period | 2.0 ms | |
| $V_{reset}$ | Reset potential | −70.0 mV | |
| $Ex_{in}$ | Stimulation | [11.9, pA 18.5, pA] | 0.14 pA |
| **Short term depression** | | | |
| Synapse Model | Depressing synapse | `tsodyks2_synapse` | |
| $\tau_{rec}$ | Time constant of recovery from STD | [300 ms, 600 ms] | 10 ms |
| $\tau_{syn}^{in}$ | Inhibitory synaptic time constant | [5 ms, 45 ms] | 10 ms |
| **Inhibitory interneurons** | | | |
| Synapse model | Static synapse model | `static_synapse` | |
| $\tau_{syn}^{ex}$ | Excitatory synaptic decay time constant | [1 ms, 101 ms] | 10 ms |
| $w_{ex}$ | Excitatory Weight | [0.1, 9.1] | 0.1 |
| $\tau_{syn}^{in}$ | Inhibitory synaptic decay time constant | 30 ms | |
| $w_{Inh}$ | Inhibitory Weight | −10 | |

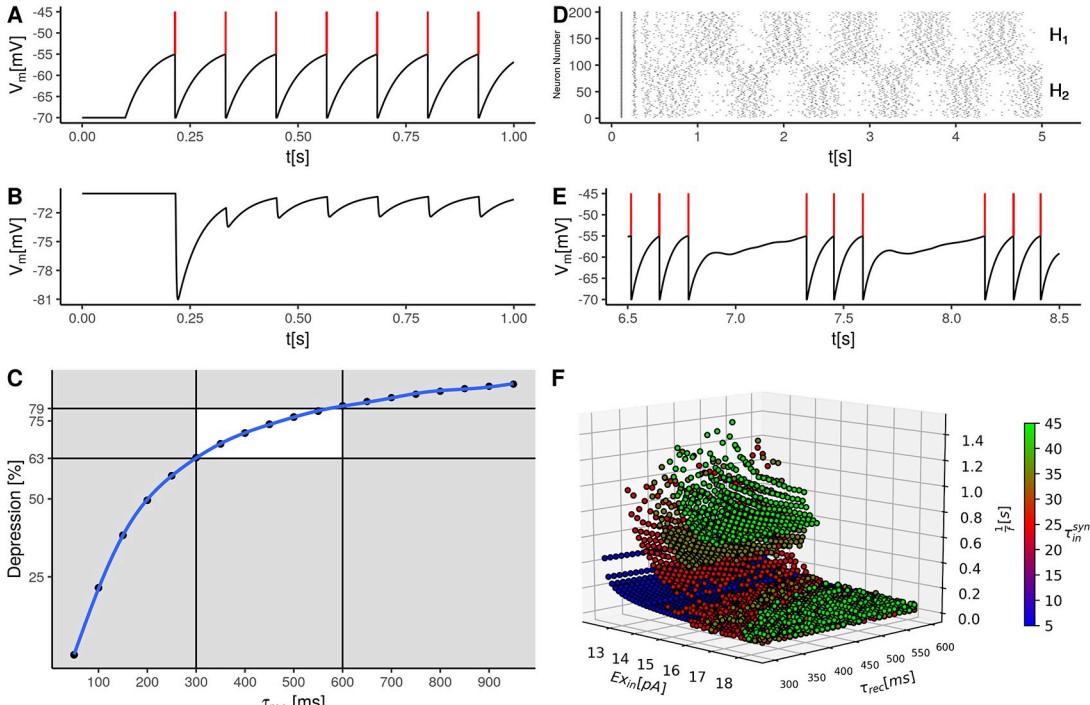

**Fig 2. Short term depression (STD) as a self-inhibitory mechanism in the half center model.** First the procedure of fitting the time constant of recovery from STD $\tau_{rec}$ is illustrated. For this purpose simulations separate from the half center model are conducted. One neuron is connected with an inhibitory depressing synapse to another neuron. (**A**) shows an exemplary voltage trace from the presynaptic neurons, which is stimulated to spike at approximately 10 Hz. Spikes are indicated in red. (**B**) shows a voltage trace from the postsynaptic neuron, Each spike shown in (**A**) generates a postsynaptic potential in the voltage trace shown in (**B**) of decreasing amplitude due to STD. (**C**) The amount of STD as a function of the time constant of recovery from STD. For each specific time constant a certain level of depression is observed in the simulation. Depressions between 63% and 79% are observed in experiments (white box area). The corresponding time constants covered an interval from 300 ms and 600 ms. (**D**) Simulating the half center model as shown in Fig 1, where the synapses connecting the half centers are subject to STD. Each half center ($H1$ and $H2$) contains one hundred neurons. The spike train of each neuron is plotted in one horizontal line. In this particular example the half centers show alternating oscillation. (**E**) Voltage trace from an exemplary single neuron from one half center whose spike trace is shown in (**D**). Here between the phases of spiking activity the depressing inhibition from the opposing half center is visible. (**F**) The observed period lengths of the oscillations in the explored parameter range are shown. Each point in the plot represents one set of parameters where oscillation occurred at a certain period length $\frac{1}{f}$ and stimulation intensity $Ex_{in}$ and time constant of recovery from STD $\tau_{rec}$. The coloring indicates the inhibitory synaptic decay time constant $\tau_{in}^{syn}$.

first inhibiting the second. The first neuron is stimulated by injection of constant current to make it spike at a frequency of approximately 10 Hz (Fig 2A). By setting the inhibitory synaptic decay time constant to 2 ms, the postsynaptic potentials induced by each presynaptic spike decay nearly to resting potential (Fig 2B). We then simulated with a broad range of values for $\tau_{rec}$. In our simulations at $\tau_{rec}$ = 300 ms a depression of approximately 63% is observed and at $\tau_{rec}$ = 600 ms a depression of 79% (Fig 2C). We use these depressions as cutoffs for possible values of $\tau_{rec}$.

We next explored the frequency of the alternating oscillations obtained in the half center model within this range. The simulation consisted of injected excitation $Ex_{in}$ into both $H_1$ and $H_2$, where each of the two half centers were simulated to contain (N = 100) neurons, as shown in Fig 1. One of the half centers by competition took the lead in the first oscillation cycle and inhibited the output of the other. Due to STD in the inhibitory synaptic influence, however, the leading half center soon lost its upper hand whereby the other half center became more excited and instead started to suppress the output of the first half center (Fig 2D). This is

shown in greater detail in Fig 2E, where decreasing inhibition due to STD on one single neuron is visible.

The generated frequencies critically depend on the synaptic decay time constant $\tau_{in}^{syn}$ and the time constant of recovery from STD $\tau_{rec}$, see Fig 2F. The longer the synaptic decay time constant the longer the possible period lengths. As shown in Fig 2F (green data points), only very long time constants were successful in achieving slow frequencies of oscillation. This is an important observation since time constants *in vivo* are much smaller than *in vitro*, see Discussion. For shorter synaptic decay time constants (blue data points in Fig 2F), expected to be more relevant in the adult spinal cord *in vivo*, oscillation frequencies below 5 Hz are observed only close to threshold activation. This observation is important because close to threshold activation, the spike generation of spinal interneurons *in vivo* becomes highly unreliable [28] and it is hence questionable whether this mechanism would at all work *in vivo*. Another caveat with this network construct is that at high stimulation intensities, the synapses become more depressed which decreases the effectiveness of the inhibition of the opposing half center. At this point the oscillations become irregular, which can be seen as dramatic drops in the period length. For example a drop from approximately 0.7 s to approximately 0.1 s for the green data points across all recovery time constants Fig 2F. Such drops merely indicate that the oscillation frequency became irregular. Hence, in settings of the synaptic decay time constant for which oscillation frequencies down to 1 Hz can be obtained, there is instead a large gap of frequencies over which no reliable control can be achieved. Hence, in summary, there are a number of short-comings in this network model which makes it unlikely as the mechanism that can deliver locomotion frequencies across the entire range observed *in vivo*.

## Self inhibition via inhibitory interneurons

A self-inhibitory mechanism that allows alternating oscillations between two groups of neurons could also be formed by a neural circuit, which does not rely on synaptic STD. Instead it uses some form of network accumulator to govern the delay of the self-inhibition so that a range of oscillation frequencies can be produced. In its simplest form, each half center has an excitatory connection to an external inhibitory neuron pool (Fig 3A). Note that each half center similar to the simulated network above, is here modeled as a group of neurons, whereas each half center is symbolized with a single neuron in Fig 3A. The half centers $H_1$ and $H_2$ provide inhibitory synapses to each other and excitatory synapses to their respective external pool of inhibitory interneurons which subserve the function of the accumulator of excitation. Hence, $H_1$ and $H_2$ are in this case simulated as containing both excitatory and inhibitory neurons.

Models with similar components as described here have been investigated for their capability to generate oscillation. In one such study, a model of spinal neuronal connectivity containing inhibitory and excitatory neurons receive external input from a CPG [50], which in that model is responsible for the rhythm generation. The mechanism of rhythm generation in the CPG is not explored. Another study [51] describes a half center model where each center also contains inhibitory and excitatory neurons and the neurons are equipped with a slow intrinsic current, which is the self-inhibitory mechanism responsible for rhythm generation of this model. The model considered here solves the self-inhibition mechanism for rhythm generation in a different way. When one half center is active, the excitation level builds up in the external pool of inhibitory neurons until it becomes sufficiently active to inhibit the active half center. When this happens, the activity shifts to the previously inactive center due to disinhibition between $H_1$ and $H_2$ (Fig 3A).

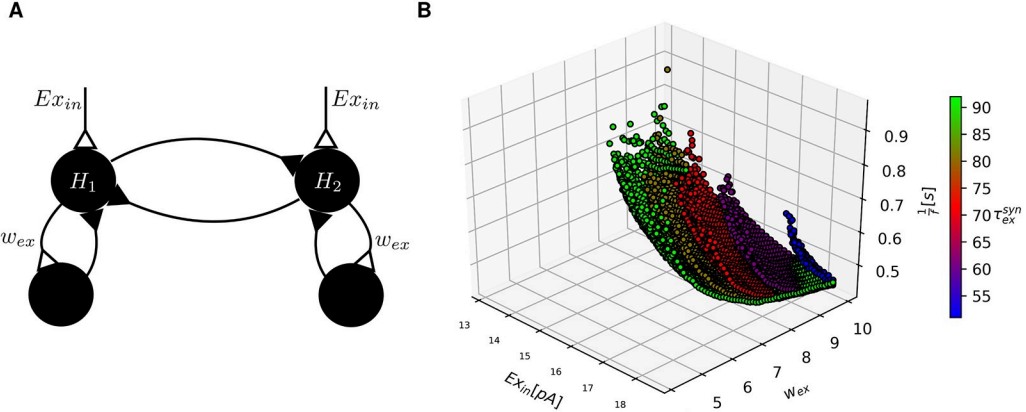

**Fig 3. The half center model with self-inhibition via auxiliary inhibitory neural circuits is simulated.** (**A**) Two half centers $H_1$ and $H_2$ are connected reciprocally with inhibitory synapses, as in Fig 1. In addition, each half center excites an external pool of inhibitory neurons. This pool in turn inhibits the half center that excites it. (**B**) Each point in the plot represents one specific combination of parameter settings ($Ex_{in}$ and the weight of the excitatory synaptic connection from the half center to the external interneurons $w_{ex}$) and the frequency of oscillation $f$ obtained. The coloring of the points indicates the excitatory synaptic decay time constant $\tau_{ex}^{syn}$.

In the here considered model the exact value of the synaptic decay time constant of inhibition is not critical, as the only determinant of the oscillatory behavior is the decay time constant of the excitatory synapses and their weight. For this mechanism to generate alternating oscillation at frequencies that would be suitable for locomotion, it is necessary that the excitation in the external inhibitory pool accumulates slowly. This requires a small increase of depolarization in the external pool with each spike that is elicited from the half center, which in principle is obtained by temporal summation of consecutive synaptic responses. This mechanism also requires a slow decay of this depolarization, which is also present in the form of the decay time constant of the synaptic response of the neuron. The time constants of synaptic decay for excitatory synapses at spinal interneurons are upper bounded by 100 ms [38–40]. This also limits the range of oscillations that can be generated with this mechanism, as illustrated in Fig 3B. Simulating this mechanism of self-inhibition over a large range of parameters, we observed oscillations in the range relevant for locomotion from 1 Hz to 10 Hz. The firing frequency of the model neurons when stimulated constantly has a strict lower bound. It would be conceivable that a neuron model with an even lower possible firing frequency could generate slower oscillations. However the same effect can be achieved by lowering the synaptic weights from the half center to its external pool of inhibitory neurons. Thus using a neuron model with a lower possible firing frequency would simply correspond to a relabeling of the stimulation intensity and synaptic weight axis in Fig 3B. Hence this self-inhibitory mechanism could make a CPG suitable for locomotion work. However similar to the results from the simulation with short term depression, the observable frequencies critically depend on the time constants. Here at excitatory synaptic decay time constants below 50 ms we did not observe any oscillations. This is an important observation, since it is unlikely that the synaptic decay time constants of spinal interneurons in the adult mammalian spinal cord *in vivo* are higher than 50 ms, see Discussion. For slow oscillation even higher synaptic decay time constants are necessary. This puts also this mechanism in question if it could make a locomotor CPG work in the adult mammalian spinal cord.

## Discussion

We have shown that self-inhibition is necessary for pattern generation in the half center hypothesis of central pattern generation for locomotion. As described in the introduction, persistent inward currents, which are a commonly used mechanism to provide the necessary delayed self-inhibition in present day computational models of the CPG, do not appear to be present in the neurons of the adult spinal cord *in vivo*. We have outlined in the introduction the underlying neuronal mechanisms and support for this observation based on direct recordings *in vivo*. Therefore, we tested the viability of two fundamental network mechanisms of delayed self-inhibition, mutual inhibition with synaptic short term depression and mutual inhibition with an accessory accumulator network, as potentially suitable candidate mechanisms to generate the alternating oscillations across the range of frequencies required for the half center model to account for the range of observed locomotion frequencies. The results show that whereas both mechanisms in principle can generate alternating oscillations across a range of frequencies, it is questionable if either one of them could sustain the full range of locomotion frequencies at any given parameter setting. Moreover, as described below, the synaptic decay time constants, at which the widest range of output frequencies were obtained, were so high compared to known synaptic decay time constants of other synapses *in vivo* that it is questionable whether these time constants could be present in the adult mammalian spinal cord *in vivo*.

For self-inhibition via inhibitory interneurons, the oscillation frequencies which can be achieved critically depend on the time constants of synaptic decay. Very high time constants are required to achieve the entire frequency range or even oscillation at all. *In vivo* synaptic decay time constants are known to be much smaller than *in vitro*. In fact, across a large number of brain structures and neuron sizes, synaptic decay time constants in adult animals *in vivo* are typically below 10 ms and at any rate appear to always be below 20 ms [34, 52–56]. This is substantially lower than the time constants that we needed to make the studied network structures to be able to generate alternating oscillatory output across the 1 Hz to 10 Hz frequency range (Figs 2 and 3).

The data of the study used for simulating the time course of the STD [36] was obtained *in vitro*. It is likely that the time constants of recovery from STD *in vivo* are much faster, possibly by a factor ten or more, as has been shown for cerebellar granule cells *in vivo*, compared to the same cells *in vitro* [52]. Hence, even though the STD in the present simulations actually could be used to obtain oscillations with as long periods as about 1.5 s, it is likely that this would not have been possible with time courses of STD that apply to the adult spinal cord *in vivo*, had such recordings been available. Furthermore *in vitro* studies show that when blocking inhibitory synapses the CPG loses its right left alternation but not its intrinsic oscillatory activity [21]. Both self-inhibitory mechanisms we proposed require inhibitory synapses. This could complicate efforts to identify these mechanisms *in vivo*.

It is possible that there exists a self-inhibitory mechanism with a variable delay that we did not consider. However in order to prove that there exists a CPG, a self-inhibitory mechanism that could be operative in the spinal cord in an adult animal *in vivo* remains to be demonstrated. This mechanism must be able to generate the entire range of locomotion, where we found that low frequencies were especially difficult to support with the explored mechanisms. Hence it is worth to discuss new explanations of how locomotion is generated that do not require a central pattern generator. In the same context it is worth to discuss how presumed observations of CPGs could be explained and how these relate to the control of locomotion.

We identify four categories of experiments in which evidence compatible with CPGs is observed. These are fictive locomotion in *in vitro* preparations, locomotion of decerebrated or

spinalized animals, ablation of neurons presumed to be involved in the CPG and fictive loco-motion in *in vivo* preparations.

For *in vitro* preparations, persistent inward currents have been shown to exist. Hence those can explain fictive locomotion *in vitro*. However as described in the introduction, in the *in vivo* state currently available data indicates that this type of mechanism cannot support fictive locomotion.

Decerebrated animals have been found to the able to walk on propelled treadmills [1, 2]. Also spinally injured animals have been found to regain their ability to walk [57, 58]. But in neither case do these findings require a CPG to be explained. Instead, an equally viable expla-nation for locomotion in decerebrated or spinally transected animals is that the biomechanics of the bodies of quadrupeds and bipeds [59–62] are naturally inclined to locomote. These stud-ies show that it is possible to construct pure mechanical systems, called passive dynamic walk-ers, with striking similarities to the mechanics of animals, which are able to locomote by only the application of an external force, while generating locomotion patterns very similar to that of animals. In the case of animals regaining their ability to walk after spinal cord injury, an additional explanation could be that the spinal network structure after developmental learning is entrained to, at least under some states of excitability, to create such neural output patterns that would be reminiscent of a CPG. In such case, a variety of non specific excitatory input to the spinal cord would be expected to create the types of neural activation patterns that we auto-matically identify as being compatible with CPGs. Long term synaptic plasticity in the spinal cord has been demonstrated [63–65]. The spinal cord is also able to adapt to produce locomo-tion despite significant experimental changes in muscle insertion sites [66]. Models of the spi-nal cord with the application of biological plausible learning [67] have been demonstrated. Whereas our study seems to exclude that such output patterns could be generated based on local connectivity patterns in the adult spinal cord, it is well known from the field of artificial recurrent neural networks that large networks, which could for example correspond to the entire spinal cord circuitry, are capable of learning and encoding almost any sequence of out-put patterns [68]. Modelling studies show that a pure reflex based model is sufficient to control locomotion [69].

Genetic ablation and inactivation experiments [37, 70–74] implicate certain neurons to be part of the CPG. These experiments make the case, that if the animal loses some of its locomo-tor functionality when one neuron type is ablated, it must be part of the CPG. However, imag-ine the case that the entire spinal cord circuitry is gradually entrained to locomotor rhythms and in response adapts its overall circuitry structure during development. By ablating neurons from birth, one would hamper the ability of the spinal cord to learn and acquire such a net-work structure. In the case of acute inactivation, as with optogenetic methods, observed effects may instead be due to that the inactivation introduces imbalances of excitability that in a heavily interconnected network may cause widespread malfunction. This is not the same as pinpointing the inactivated neurons as being solely responsible for producing crucial aspects of the output pattern. Similar to when animals regain their ability to walk after spinal cord injury, the spinal cord could have been entrained to locomote before the injury. The entrained network is then simply reactivated after the injury. Implicating certain neurons in the CPG by ablation and inactivation experiments requires the assumption that a CPG exists. Experiments with spinal injury experiments fall under this same umbrella. Hence genetic ablation, inactiva-tion and spinal cord injury experiments are not a proof of the existence of a CPG.

The most compelling proof for a CPG are experiments that demonstrate fictive locomotion *in vivo* in adult animals. In one particular preparation [75] an adult animal produces fictive locomotion after decerebration and after spinalization, while the spinal cord is still attached to the body, but is disconnected from modulating its sensor feedback due to pharmacological

muscle paralysis or transaction of efferent motor nerves. There 5-HT, L-DOPA and a mono-
amine oxidase inhibitor are required to elicit fictive locomotion. The nature of these experi-
mental conditions make it impossible to exclude that the 5-HT and dopamine concentrations
reach unphysiological high levels under which the efficacy of ion channels brings the neurons
into an unnatural state in which they are suddenly likely to be capable of pacemaking-like
activity. The neuromodulator 5-HT promotes persistent inward currents [76], so that even in
an *in vivo* experiment the neurons could be made to behave like in an *in vitro* setting, where
intrinsic conductances become more dominant than the total synaptic input activity to the
neuron. In another set of preparations [29, 77] the animal was also paralyzed and fictive loco-
motion was evoked by stimulation of the mesencephalic locomotor region. In this case it can-
not be excluded that the oscillatory behavior in the spinal neurons is due to the descending
activation from the brainstem. For example the serotonergic neurons of the brainstem contain
pacemaker neurons [78, 79] which have connections to the spinal cord and whose pacemaking
activity could be responsible for the necessary self-inhibition underlying the alternating oscilla-
tions that were measured.

Hence a CPG is not the only possible explanation for real locomotion in intact animals and
the presumed CPG observations in non intact preparation, many times in newborn or juvenile
animals may not generalize to the intact adult mammal. The possibilities include that the bio-
mechanics of the bodies of quadrupeds and bipeds [59–62] have a natural inclination to gener-
ate locomotion automatically and that the structure of the spinal cord circuitry gradually
becomes entrained to patterns of neural activity that are associated with those locomotion
patterns.

## Methods

First the general models and tools are described, then we describe the simulations we per-
formed in detail. The goal of the simulations are to investigate if the time scale of the self-
inhibitory mechanisms, short term depression, and self-inhibition via inhibitory interneurons
are suitable to generate oscillations for locomotion.

### Simulation tools

For simulation we used NEST [80]. We did use the git revision from Feb 13 2018 available at
https://github.com/nest/nest-simulator. For processing network models we used graph-
tool [81]. Curve fitting for oscillation detection was done using least squares from SciPy
[82].

### Neural models and parameters

For all simulations we used NESTs implementation of leaky integrate and fire neurons with
alpha function shaped synaptic currents. All model parameters are listed in Table 1. Spike
threshold, reset potential, refractory period and minimum membrane potential were left at the
default values of NEST. For the membrane capacitance and the membrane time constant used
values measured in spinal interneurons [37]. Further parameters for the neuron model are the
excitatory and inhibitory synaptic decay time constants $\tau_{in}^{syn}$ and $\tau_{ex}^{syn}$. These are set to different
values in the simulations based on experimental data.

### Synapse models and parameters

We used three different synapse models, which are available in NEST. As model of synapses
subject to short term depression we used the model tsodyks2_synapse, which simply

scales the weight of the synapse [47–49]. The static synapse model `static_synapse` was used for simulations with self-inhibition via inhibitory interneurons. The synaptic delay was set to 1 s in all models [83, 84].

## Network structure

For all simulations, we used a common neuronal network that can be separated into four pools of neurons. Each pool consists of one hundred neurons. Two pools represent the half centers of the CPG. The two other pools represent the additional inhibitory pools for the self-inhibition via inhibitory interneurons mechanism, which were only present in the simulations that use inhibitory interneurons as the self-inhibitory mechanism. Two neuron pools were connected at random by inserting a connection with probability 0.1 for each pair of neurons. The pools representing the half centers were connected as displayed in Fig 1, where each node in the figure corresponds to one pool. For simulations with the self-inhibition via inhibitory interneurons, each half center was connected to the additional inhibitory pool with excitatory connections. The inhibitory pool in turn then was connected with inhibitory connections back to the half center.

## Simulation procedure

In order to determine if short term depression and self-inhibition via inhibitory interneurons are suitable self-inhibitory mechanism in the half center model we scanned over a wide range of parameters. The parameters, their values and step sizes for the scan are listed in Table 1. For each set of parameters the half centers were simulated for 10*s* and it was determined if oscillation emerge, as described below. In Figs 2F and 3B each point in the plot represents a set of parameters that generated oscillation. Sets of parameters that did not generate oscillation are not plotted.

## Oscillation detection and frequency estimation

In order to automatically determine if alternating oscillation between the half centers exists in one particular simulation, we used the following procedure. First we computed the firing rate of each single neuron, which is the low pass filtered version of the spike train [42]. Let $t_1, \ldots t_m$ be the sequence of spike times of the neuron $n$ and $\tau$ the time constant of the low pass filter. The firing rate $v_n(t)$ is defined as

$$\tau \frac{dv_n(t)}{dt} = -v_n(t) + \sum_{1 \leq i \leq m} \delta(t - t_i) \ ,$$

where $\delta$ is the Dirac delta function. We set $\tau = 100$ ms, which is a compromise between detecting high frequency oscillations and smoothing the neurons spikes. Let $v_{i,1}(t)$ be the firing rate of neuron $i$ in the first half center and $v_{i,2}(t)$ the firing rate of neuron $i$ in the second half center. The firing rate $v_{\mathrm{CPG}}$ of the CPG is defined as

$$v_{\mathrm{CPG}}(t) = \frac{1}{n} \sum_{1 \leq i \leq l} v_{i,1}(t) - \frac{1}{n} \sum_{1 \leq i \leq l} v_{i,2}(t) \ ,$$

i.e. the difference of average firing rates of the half centers. Firing rates are computed with Euler Integration with a time step of 0.1 ms. Let $v_{\mathrm{CPG}}(t)$ be the solution of the integration. We fit a sine function to the firing, parameterized by an amplitude $B$, a frequency $f$ an offset in

time $\Delta t$ and an offset in firing rate $\Delta v$ by solving the minimization problem

$$\min \|\bar{v}_{\text{CPG}}(t) - B\sin(2\pi ft + \Delta t) - \Delta v\|_2^2 \;.$$

If $B \geq 2\Delta v$, i.e. when the sine function oscillates strongly around zero, the simulation, respectively the parameter set is classified as oscillatory. In this case the frequency of the pattern is estimated as $f$.

Solving the optimization problem requires a suitable set of initial values for the parameters. The offset parameter $\Delta v$ is set to the average of $v_{\text{CPG}}(t)$ and the offset $\Delta t$ is set to zero. The amplitude $A$ is estimated as the maximum of $v_{\text{CPG}}(t)$ over all $t$. To initialize the frequency $f$ the autocorrelation of $v_{\text{CPG}}(t)$ is computed. Let $t'$ be the position of the second largest local maximum of the autocorrelation. The frequency is set to $f = \frac{1}{t'}$.

This method classified some parameter sets as oscillatory, which are instances when there is zero firing rate, or there is some oscillation which dies down after a few cycles or oscillation with totally irregular intervals. Oscillations with greatly irregular intervals is unsuitable for locomotion. All of these missclassifications are characterized by the method estimating a very low frequency, which greatly deviates from the frequencies estimated of instances with very close parameters. We reclassified those instances manually to not oscillatory.

## Author Contributions

**Conceptualization:** Matthias Kohler, Philipp Stratmann, Florian Röhrbein, Alois Knoll, Alin Albu-Schäffer, Henrik Jörntell.

**Formal analysis:** Matthias Kohler, Philipp Stratmann, Henrik Jörntell.

**Writing – original draft:** Matthias Kohler, Henrik Jörntell.

**Writing – review & editing:** Matthias Kohler, Alin Albu-Schäffer, Henrik Jörntell.

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
