## [Decision Letter · Decision Letter 0]

6 Jul 2020

PONE-D-20-07412

Biological data questions the support of the self inhibition required for pattern generation in the half center model.

PLOS ONE

Dear Dr. Kohler,

Thank you for submitting your manuscript to PLOS ONE. After careful consideration, we feel that it has merit but does not fully meet PLOS ONE’s publication criteria as it currently stands. Therefore, we invite you to submit a revised version of the manuscript that addresses the points raised during the review process.

We look forward to receiving your revised manuscript.

Kind regards,

Seil Sohn, MD,PhD

Academic Editor

PLOS ONE

Journal Requirements:

Additional Editor Comments:

This is resubmission paper, which descirbied half center model with self inhibition concept by simulation method. Authors made greatly efforts and I appreciate your efford in this field. It was very hard to recruite proper expert reviewers, but one reviewer thankfully completed comments. I think that several concerens should be addressed in order to be published in this journal.

Reviewers' comments:

Reviewer's Responses to Questions

**Comments to the Author**

1. Is the manuscript technically sound, and do the data support the conclusions?

Reviewer #1: Partly

2. Has the statistical analysis been performed appropriately and rigorously? 

Reviewer #1: Yes

3. Have the authors made all data underlying the findings in their manuscript fully available?

Reviewer #1: Yes

4. Is the manuscript presented in an intelligible fashion and written in standard English?

Reviewer #1: Yes

5. Review Comments to the Author

Reviewer #1: Authors propose a notion that the half center model with self inhibition concept is the better model for CPG. This concept contributes to the development of robotics field such as walking machinery.

However, from the point of view of electrophysiology, assumption that interneurons and/or inhibitory neurons is just inhibitory block (one parameter settings) seems to be too simple.

Bursting could cause to increase the rise of intracellular Cl- concentrations. Eventually high concentrations of intracellular Cl- switch inhibitory GABAergic and glycinergic inputs to excitatory ones. Generally, frequency of firing of inhibitory neurons is much higher than that of excitatory neurons. In addition, intracellular Cl- homeostasis regulated by Cl- transporters is quite difference between juvenile and adult animals.

6. PLOS authors have the option to publish the peer review history of their article (what does this mean?). If published, this will include your full peer review and any attached files.

Reviewer #1: No

---

## [Author Response · Author response to Decision Letter 0]

7 Aug 2020

# Response to the reviewer comments for submission PONE-D-20-07412

> Reviewer #1: Authors propose a notion that the half center model with self

> inhibition concept is the better model for CPG. This concept contributes to the

> development of robotics field such as walking machinery. However, from the

> point of view of electrophysiology, assumption that interneurons and/or

> inhibitory neurons is just inhibitory block (one parameter settings) seems to

> be too simple. Bursting could cause to increase the rise of intracellular Cl-

> concentrations. Eventually high concentrations of intracellular Cl- switch

> inhibitory GABAergic and glycinergic inputs to excitatory ones. Generally,

> frequency of firing of inhibitory neurons is much higher than that of

> excitatory neurons. In addition, intracellular Cl- homeostasis regulated by Cl-

> transporters is quite difference between juvenile and adult animals.

The process described by the reviewer could indeed provide an additional self

inhibitory mechanism that we did not consider in the paper. The reviewer

correctly points out that differences between juvenile and adult neuron

physiology exist. In the adult a switch from inhibition to excitation does not

seem to happen. Evidence for that exists from recordings in the spinal cord of

adult cats, where the action of inhibitory neurons on motoneurons and

interneurons was studied (Hultborn Jankowska Lindström 1971, Jankowska Roberts

1972). There high frequency stimulation evokes inhibitory postsynaptic

potentials in the recorded neurons. However a switch from inhibition

to excitation is not observed. Hence for the adult, which our manuscript is

focused on, based on the available data, it is reasonable to assume that

inhibition does not switch to excitation. We have updated the manuscript to

to justify the usage of inhibition which can be reliably maintained.

For the juvenile, which our paper is not focused on, the situation is more

confusing. It has been found that Gaba acts excitatory in juvenile neurons

(Ben-Ari 2012). However counterevidence exists which shows that Gaba acts

inhibitory in juvenile neurons (Kirmse et al. 2015, Valeeva et al. 2016). 

During epilepsy GABA might have excitatory effects in the

brain (Kaila et.al. 2014). If such effects are a potential cause for seizures,

it is unlikely that they would be involved during locomotion.

## References

Hultborn Jankowska Lindström 1971,

Recurrent inhibition of interneurones monosynaptically activated from group Ia afferents

<https://doi.org/10.1113/jphysiol.1971.sp009488>

Jankowska Roberts 1972,

Synaptic actions of single interneurones mediating reciprocal Ia inhibition of motoneurones

<https://dx.doi.org/10.1113%2Fjphysiol.1972.sp009818>

Ben-Ari 2012,

The GABA excitatory/inhibitory Shift in Brain Maturation and Neurological Disorders 

<https://doi.org/10.1177/1073858412438697>

Kirmse et al. 2015,

GABA depolarizes immature neurons and inhibits network activity in the neonatal neocortex in vivo

<https://doi.org/10.1038/ncomms8750>

Valeeva et al. 2016,

An Optogenetic Approach for Investigation of Excitatory and Inhibitory Network GABA

Actions in Mice Expressing Channelrhodopsin-2 in GABAergic Neurons 

<https://doi.org/10.1523/jneurosci.3482-15.2016>

Kaila et.al. 2014

GABA actions and ionic plasticity in epilepsy

<https://doi.org/10.1016/j.conb.2013.11.004>

---

## [Decision Letter · Decision Letter 1]

20 Aug 2020

Biological data questions the support of the self inhibition required for pattern generation in the half center model.

PONE-D-20-07412R1

Dear Dr. Kohler

We’re pleased to inform you that your manuscript has been judged scientifically suitable for publication and will be formally accepted for publication once it meets all outstanding technical requirements.

Kind regards,

Seil Sohn, MD,PhD

Academic Editor

PLOS ONE

Additional Editor Comments (optional):

Thanks for your effort.

Reviewers' comments:

Reviewer's Responses to Questions

**Comments to the Author**

1. If the authors have adequately addressed your comments raised in a previous round of review and you feel that this manuscript is now acceptable for publication, you may indicate that here to bypass the “Comments to the Author” section, enter your conflict of interest statement in the “Confidential to Editor” section, and submit your "Accept" recommendation.

Reviewer #1: All comments have been addressed

2. Is the manuscript technically sound, and do the data support the conclusions?

Reviewer #1: Yes

3. Has the statistical analysis been performed appropriately and rigorously? 

Reviewer #1: Yes

4. Have the authors made all data underlying the findings in their manuscript fully available?

Reviewer #1: Yes

5. Is the manuscript presented in an intelligible fashion and written in standard English?

Reviewer #1: Yes

6. Review Comments to the Author

Reviewer #1: (No Response)

7. PLOS authors have the option to publish the peer review history of their article (what does this mean?). If published, this will include your full peer review and any attached files.

---

## [Editor Report · Acceptance letter]

28 Aug 2020

PONE-D-20-07412R1 

Biological data questions the support of the self inhibition required for pattern generation in the half center model. 

Dear Dr. Kohler:

I'm pleased to inform you that your manuscript has been deemed suitable for publication in PLOS ONE. Congratulations! Your manuscript is now with our production department. 

Kind regards, 

on behalf of

Dr. Seil Sohn 

Academic Editor

PLOS ONE